# Vitamin C Plasma Levels Associated with Inflammatory Biomarkers, CRP and RDW: Results from the NHANES 2003–2006 Surveys

**DOI:** 10.3390/nu14061254

**Published:** 2022-03-16

**Authors:** Jennifer Marie Crook, Ann L. Horgas, Saunjoo L. Yoon, Oliver Grundmann, Versie Johnson-Mallard

**Affiliations:** 1Center for Health Equity and Community Engagement Research, Mayo Clinic Florida, Jacksonville, FL 32224, USA; 2Biobehavioral Nursing Science, College of Nursing, University of Florida, Gainesville, FL 32610, USA; ahorgas@ufl.edu (A.L.H.); yoon@ufl.edu (S.L.Y.); 3Entrepreneurial Programs in Medicinal Chemistry, College of Pharmacy, University of Florida, Gainesville, FL 32610, USA; grundman@ufl.edu; 4College of Nursing, Kent State University, Kent, OH 44240, USA; vjohns29@kent.edu

**Keywords:** ascorbate, vitamin C, inflammation, CRP, RDW

## Abstract

Although undisputed for its anti-inflammatory and immune system boosting properties, vitamin C remains an inconsistently investigated nutrient in the United States. However, subclinical inadequacies may partly explain increased inflammation and decreased immune function within the population. This secondary analysis cross-sectional study used the 2003–2006 NHANES surveys to identify more clearly the association between plasma vitamin C and clinical biomarkers of acute and chronic inflammation C-reactive protein (CRP) and red cell distribution width (RDW). From plasma vitamin C levels separated into five defined categories (deficiency, hypovitaminosis, inadequate, adequate, and saturating), ANOVA tests identified significant differences in means in all insufficient vitamin C categories (deficiency, hypovitaminosis, and inadequate) and both CRP and RDW in 7607 study participants. There were also statistically significant differences in means between sufficient plasma vitamin C levels (adequate and saturating categories) and CRP. Significant differences were not identified between adequate and saturating plasma vitamin C levels and RDW. Although inadequate levels of vitamin C may not exhibit overt signs or symptoms of deficiency, differences in mean levels identified between inflammatory biomarkers suggest a closer examination of those considered at risk for inflammatory-driven diseases. Likewise, the subclinical levels of inflammation presented in this study provide evidence to support ranges for further clinical inflammation surveillance.

## 1. Introduction

As a water-soluble vitamin that humans cannot endogenously synthesize, bodily vitamin C levels rely on a consistent dietary intake of various fruits and vegetables. It is a well-known antioxidant with multiple beneficial properties in the immune system including recycling and enhancing the bioavailability of other vitamins and minerals [1], influencing DNA and histone demethylation and enzyme-dependent biosynthesis of various biomolecules [2,3] alleviating inflammation [4], and stabilizing the structure of collagen [5,6]. Vitamin C has shown potential effectiveness in the treatment of severe coronavirus disease 2019 (COVID-19) and sepsis when given to patients intravenously in the intensive care setting [7,8,9]. Although the pharmacokinetics and dosage differ with the route given, there continues to be constant elucidation of the beneficial properties of vitamin C in various bodily functions.

Multiple studies have identified significant downward trends in plasma vitamin C in critically ill patients, exclusive of decreases in other known anti-inflammatory vitamins and minerals [10,11,12]. In the recent COVID-19 pandemic, severely symptomatic hospitalized patients with COVID-19 exhibited extremely low (<25 μmol/L) plasma vitamin C levels [7] in the presence of increased inflammation.

However, patients undergoing routine annual exams or admission to inpatient settings are not normally assessed for vitamin C status, and patient reports of dietary intake may not be an adequate enough assessment [13]. This is unfortunate as research indicates not only is there a high percentage of the U.S. population at-risk for insufficient plasma levels of vitamin C (<50 μmol/L) [13], but if plasma levels drop to the range of 11–23 μmol/L (defined as hypovitaminosis), increased supplementation is needed to reverse hypovitaminosis and achieve the saturating levels necessary for maximum immune support [14,15].

It is possible that subclinical nutritional insufficiencies of vitamin C partly explain increased amounts of both acute and chronic inflammation, as well as inadequate immune responses responsible for disease instigation and progression. Our previous work examined the range of plasma vitamin C levels in five clearly defined categories (deficiency, hypovitaminosis, inadequate, adequate, and saturating) within the population of the United States [13]. There currently remains little information regarding the prevalence of inadequate plasma vitamin C (defined as the subclinical range existing between hypovitaminosis and adequate levels which may not present with clear signs or symptoms of deficiency) and its association with inflammation. Although there has been extensive literature identifying the inverse association between vitamin C and C-reactive protein (CRP) [16,17] and other nonspecific markers of inflammation including white blood cells (WBCs) and platelets [18], recognizing biomarkers for chronic inflammatory assessment remains relatively challenging.

CRP, well-established as a strong predictor of risk of developing cardiovascular disease [19], is an acute-phase protein produced mainly by hepatocytes in response to inflammatory cascade pathway signaling [19]. Levels of highly sensitive CRP (hs-CRP) are relatively similar in men and women and average approximately 1.5 mg/L among middle-aged Americans [19]. Current guidelines by the American Heart Association indicate levels of hs-CRP between 1 and 3 mg/L as medium risk and >3 mg/L as high risk for developing cardiac disease. Although the NHANES surveys used in this study did not capture high sensitivity assays, the correlation between low-sensitive CRP (ls-CRP) used in these surveys and hs-CRP has been found to be significantly correlated [20].

RDW is a count of the variability in the size of circulating erythrocytes and can highlight a disturbance in their life cycles and homeostasis. Because red blood cells live for approximately 120 days, RDW may represent a view of inflammation over time. Once used solely for differentiating anemia diagnoses, RDW has been identified as associated with CRP, as well as multiple chronic diseases with inflammatory components [21,22,23,24]. To date, there is little evidence of a relationship between RDW and vitamin C plasma levels [25,26]. Current guidelines for normal ranges of RDW are 12.2 to 16.1% in adult females and 11.8 to 14.5% in adult males.

The purpose of this study was to investigate the relationships between five defined plasma vitamin C categories (deficiency, hypovitaminosis, inadequacy, adequacy, and saturating) and levels of acute and chronic inflammation, via the biomarkers of C-reactive protein (CRP) and red cell distribution width (RDW).

## 2. Materials and Methods

Data for this study were gleaned from the National Health and Nutrition Examination Surveys (NHANES). These nationally representative annual surveys utilize a complex, multi-stage sampling design which necessitates the proper calculation and inclusion of sampling weights when conducting analysis. Details regarding the sampling design and sample weight construction criteria can be found on the Centers for Disease and Control (CDC) website [27]. Inclusion criteria consisted of all genders and ethnicities, ages > 20 years of age, non-institutionalized civilian participants who were able to give informed consent and participated in both questionnaire and laboratory assessment measurements. Excluded from analysis were children, individuals in the military, institutionalized individuals, and participants with incomplete data from combined participant laboratory and interview portions. Mobile exam clinics were utilized for blood collection, where they were minimally processed and shipped to remote laboratories for assay processing. All variables analyzed in this study were taken from NHANES data collection which was made publicly available on their website in the 2003–2004 and 2005–2006 surveys. Blood level values used in this study were not fasting laboratory values. More detailed information regarding the laboratory collection, processing, and reporting of the NHANES survey variables used in this study can be found on the CDC website [27,28]. The sample selection pathway for this study, as well as the variable descriptions has been previously published [13]. Food security information was included as a response to the question, “Are you worried you will run out of food?”. From the NHANES surveys completed and published for the years 2003–2004 and 2005–2006, this study cohort included a final sample size of 7607 unique participants.

Plasma vitamin C (ascorbic acid) was collected and measured by isocratic high-performance liquid chromatography with electrochemical detection at 650 mV. Peak area quantitation was based on a standard curve that was generated from three different concentrations of an external standard (0.025, 0.150, and 0.500 mg/dL). The quality assurance and quality control protocols utilized by NHANES meet the 1988 Clinical Laboratory Improvement Act mandates. A full description of the specimen collection, laboratory processing method, and quality control procedures for vitamin C can be found on the CDC website [28,29]. There were correlations identified in initial analysis between the continuous vitamin C variable to other tested variables. The vitamin C variable was recoded into the following five categories: deficiency (0–10.99 μmol/L), hypovitaminosis (11–23.99 μmol/L), inadequate (24–49.99 μmol/L), adequate (50–69.99 μmol/L), and saturating (≥70 μmol/L) based on participant plasma levels. Although there are currently minimal variations in the international definitions of hypovitaminosis, inadequate, adequate, and saturating levels, the parameters for the ranges defined in this study were taken from studies examining hypovitaminosis and supplementation, as well as the saturating levels in which maximum immune support was achieved [14,15].

For processing of CRP, latex-enhanced nephelometry with particle-enhanced assays were used for quantitation. Assays were performed on a Behring Nephelometer to determine quantitative CRP levels. The primary standard used for processing was organized by Behring Diagnostics and standardized against WHO reference material [28]. More detailed laboratory processing information can be found on the CDC website [28]. The CRP was kept as a continuous variable for analytical purposes in this study.

RDW was included in the complete blood cell count (CBC) and processed via a Beckman Coulter MAXM Instrument which derives CBC parameters based on the Beckman Coulter method of counting, sizing, automatic diluting and mixing for sample processing, and utilizing a single beam photometer for hemoglobinometry [28]. More specific guidance on the processing of the CBC with differential specimens can be found on the CDC website [28]. RDW was listed as the NHANES variable LBXRDW, which provided a range of values as a percentage (10.6–26.9%). For this study, RDW was kept as a continuous variable, such as CRP, for analytical purposes.

Data from 2003–2006 NHANES datasets were downloaded in a Statistical Analysis System (SAS) transport file format version 9.4 (SAS Institute Inc., Cary, NC, USA). SAS files were converted to Statistical Package for the Social Sciences (SPSS) (IBM SPSS Statistics for Windows, Version 26.0. Armonk, NY, USA) for analysis. Four-year sample weights were calculated per the National Center for Health Statistics (NCHS) guidelines. In all statistical tests, a *p*-value of less than 0.05 was considered statistically significant.

Analysis of variance was used to test for differences in mean levels of CRP and RDW across levels of plasma vitamin C in five defined quintiles. The variables of CRP and RDW were assessed for normality and found to be negatively skewed, although with large sample sizes, violations of normality do not noticeably affect results and, thus, do not require transformations [30]. Collinearity diagnostics among continuous variables were evaluated with Pearson correlation tests, with no multicollinearity relationships identified. Normality of error terms was evaluated with Kolmogorov–Smirnov tests and violations were not observed. Homoscedasticity of error terms were assessed, and a violation of this assumption was revealed, rendering a Welch correction applied to ANOVA results.

## 3. Results

For the 7607 unique cases utilized in this study, Table 1 presents the demographic characteristics. Most participants in this study (40.1%) were middle aged (40–59 years), Non-Hispanic White (73.6%), female (51.3%), nonsmokers (70.6%), and food secure (85.9%). A large percentage of the sample (63.8%) indicated high poverty levels classified as “high poverty to income ratio (PIR)”. Mean BMI levels were 28.7 kg/m^2^ (SD = 6.44) and mean plasma vitamin C levels were 54.6 μmol/L (SD = 28.6). Participants’ inflammatory markers revealed a mean plasma CRP level of 0.48 mg/dL (SD = 0.92) and an RDW level of 12.9% (SD = 1.2).

In Table 2, the statistically significant differences in mean levels in both CRP (*F* = 19.4, *df* = 4, *p* = <0.001) and RDW (*F* = 11.2, *df* = 4, *p* = <0.001) in plasma vitamin C categories can be more clearly realized. Post hoc Bonferroni tests indicated significant between-group differences in the quintiles of plasma vitamin C.

For CRP, there were no statistically significant group differences between deficiency, hypovitaminosis, and inadequate plasma vitamin C categories. Mean levels of CRP within the insufficient plasma vitamin C categories (deficient, hypovitaminosis, and inadequate) were significantly higher than the means within the sufficient categories (adequate and saturating). Adequate and saturating vitamin C categories also revealed significant differences in mean CRP; with the vitamin C adequate category displaying higher CRP levels than those found in the saturating category. Among the quintiles of plasma vitamin C considered insufficient (deficiency, hypovitaminosis, and inadequate) mean CRP levels ranged from 0.53 mg/dL (95% CI: 0.50–0.57) to 0.67 mg/dL (95% CI: 0.54–0.80). Among the categories of plasma vitamin C considered sufficient (adequate and saturating), CRP means ranged from 0.37 mg/dL (95% CI: 0.01–0.40) to 0.45 mg/dL (95% CI: 0.42–0.49). See Figure 1.

Mean RDW values for all insufficient plasma vitamin C categories (deficiency, hypovitaminosis, and inadequate) were significantly higher than the means in the sufficient plasma vitamin C categories (adequate and saturating). There were no significant group differences of RDW means between the insufficient categories of deficiency, hypovitaminosis, and inadequate or between the sufficient categories of adequate and saturating. Mean ranges of RDW identified between the insufficient categories of vitamin C were 12.8% (95% CI: 12.7–12.8) to 12.8% (95% CI: 12.8–12.9), while the ranges identified within the sufficient categories of vitamin C were 12.9% (95% CI: 12.9–13.0) to 13% (95% CI: 12.9–13.2). See Figure 2.

## 4. Discussion

From the 2003–2004 and 2005–2006 NHANES surveys, mean plasma vitamin C was 54.4 μmol/L, defined in this study as adequate. Considering bioavailability of vitamin C may change depending on body requirements, the proximity of this value to the cutoff between adequacy and inadequacy should warrant concern from a public health surveillance perspective. Research has identified that over 25% of the U.S. population (pre-COVID-19) possessed inadequate vitamin C plasma levels with males, adults aged 20–59, Black and Mexican Americans, smokers, people with increased BMI, low-income and middle class, and food insecure individuals at highest risk [13]. These populations have been associated with increased inflammation as well [31,32,33], though they may not exhibit any signs or symptoms of vitamin C deficiency. How the recent COVID-19 pandemic and resulting quarantines, school closures, job losses, and supply shortages have affected that number is currently unknown.

This study revealed significant differences in means of both inflammatory biomarkers CRP and RDW in five plasma vitamin C quintiles. All categories of plasma vitamin C levels were associated with CRP levels of <1 mg/dL and mean RDW percentages of 12.9–13.0% which are currently considered low risk for cardiovascular disease, according to current AHA guidelines [19] and not clinically recognized as relevant. However, upon closer examination, the mean difference between insufficient categories (deficiency, hypovitaminosis, and inadequate) are significantly different than those with sufficient plasma vitamin C categories (adequate and saturating). This finding provides evidence to support a review, and revision of inflammatory surveillance ranges with consideration for differentiating between acute and chronic inflammation. In this study, insufficient quintiles of plasma vitamin C indicated mean CRP levels of >0.5 mg/dL and mean RDW levels of ≥12.9%. Although there is no currently specific guidance for RDW in inflammatory surveillance, this study provides unique insight into possible parameters for use of RDW to assess chronic inflammation. This is the first known study to correlate plasma vitamin C with RDW levels. It is also interesting to note that there were statistically significant mean differences in CRP levels between both adequate and saturating vitamin C categories, providing evidence that saturating plasma vitamin C levels may be the standard range for reduction in inflammation [15,34]. Though research has found low levels of hs-CRP to be nonspecific for inflammation [35], the differences in this study suggest that the AHA guidelines for hs-CRP of 1–3 mg/dL may need review, to ensure that they capture the issue of chronic inflammation and/or vitamin C deficiencies which are important for preventative inflammation surveillance. This is corroborated by the significant mean difference of RDW across all insufficient vitamin C categories (deficiency, hypovitaminosis, and inadequate).

Limitations of this study include the use of secondary data, which limits the ability to define research variables, revise data collection, and add variables not previously collected. Physical activity, a known component associated with vitamin C bioavailability, was not addressed in this study [36,37] due to the limited definition of physical activity captured in the NHANES surveys. Future research is suggested to include physical activity in study designs. Another limitation is the age of the data in this study. A national assessment of vitamin C was last conducted in the United States prior to the COVID-19 pandemic and a closer examination of the plasma status of the population after pandemic measures were instituted is advised. The results of this study suggest that a re-assessment of vitamin C status is warranted, as pandemic measures, such as unemployment, school closures, food shortages, and lockdowns [38], may have increased vitamin C insufficiencies. Other limitations include the lack of inclusion of inflammatory biomarkers IL-6 and TNF-α variables and other biomarkers, including hs-CRP which have been identified to be a more accurate diagnostic indicator of inflammation than CRP [39]. There is a lack of information about the vitamin C status in children and young adults <20, as well as more defined delineations within the older adulthood category. The large range of plasma vitamin C suggests supplement use (though not explored in this study and not clearly defined in these NHANES surveys) and the timing of dietary consumption of vitamin C items prior to blood collection as another limitation of this study and future research is suggested with these factors controlled. Finally, the study findings are generalizable to the U.S. population, although vitamin C insufficiencies may be a global problem and should be considered in other countries.

## 5. Conclusions

This study has confirmed the association between inadequate vitamin C and both acute and chronic inflammation. It has identified a novel association between plasma vitamin C and RDW, as well as confirmed and expanded the relationship between vitamin C and CRP. Associations with other biomarkers for acute and chronic inflammation are highly recommended as future directions of research. The relatively inexpensive CBC test containing RDW is suggested as an easier way to monitor chronic inflammation. Although mean CRP and RDW levels of the participants in this study are currently considered to be in normal ranges, significant between-group differences were identified between individuals possessing sufficient plasma vitamin C and those with insufficient levels. It is recommended that future research efforts continue to explore the relationship between inadequate vitamin C (defined in this study as plasma levels of 24–49.99 μmol/L) and inflammation, and explore the plasma vitamin C ranges to identify other significant associations with diseases that include inflammatory components. The plasma vitamin C quintiles presented in this study are suggested for use in routine and in-patient assessments, though it is possible that in hospitalized settings, a temporarily increased recommended daily allowance should be considered for times of increased bodily need. Finally, as vitamin C is solely obtained through diet, it is recommended that a more complete examination of other nutritional insufficiencies identified to be associated with inflammation is explored, and the surveillance of nutritional health (beyond patient report) be considered a priority.

## Figures and Tables

**Figure 1 nutrients-14-01254-f001:**
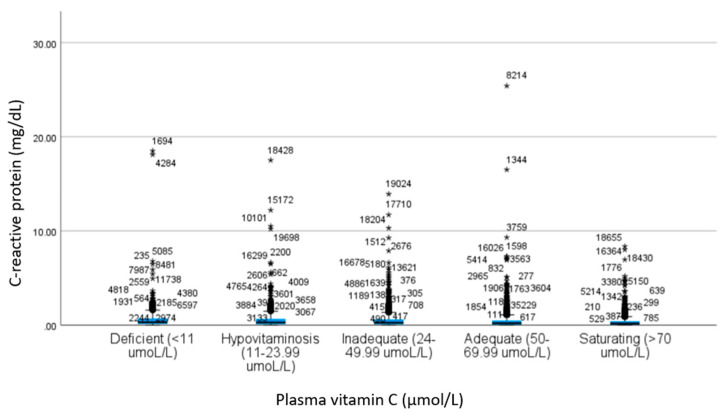
Participant CRP mean levels in plasma vitamin C categories. * denotes participant outlier.

**Figure 2 nutrients-14-01254-f002:**
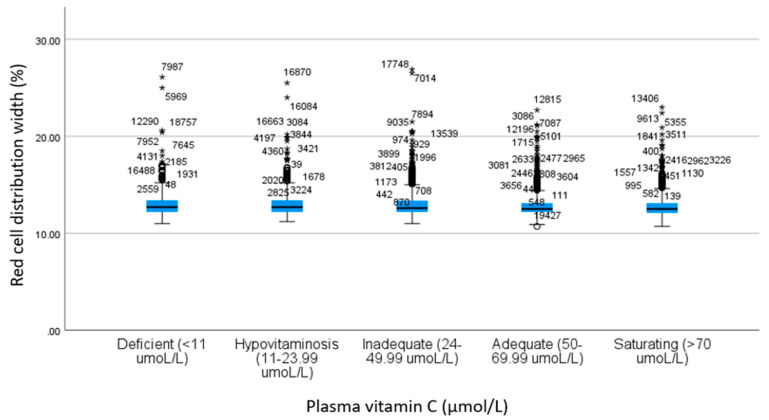
Participant RDW mean levels in plasma vitamin C categories. * denotes participant outlier.

**Table 1 nutrients-14-01254-t001:** Sample description (*n* = 7607).

Characteristics	*n*	Weighted *n* (%)	Mean (SD)	Range
Gender				
Male	3699	48.7% ± 0.7%		
Female	3908	51.3% ± 0.7%		
Age				
Young Adult 20–39	2751	37.5% ± 0.7%		
Middle Adult 40–59	2295	40.1% ± 0.7%		
Late Adult ≥ 60	2561	22.4% ± 0.5%		
Race/Ethnicity				
Mexican American	1516	7.6% ± 0.2%		
Other Hispanic	230	3.4% ± 0.3%		
Non-Hispanic White	4305	73.6% ± 0.5%		
Non-Hispanic Black	1536	10.5% ± 0.3%		
Other	290	4.9% ± 0.3%		
Family PIR ^1^				
High (0–1.5)	5206	63.9% ± 0.5%		
Medium (1.51–4.5)	1614	22.6% ± 0.5%		
Low (>4.51)	787	13.5% ± 0.5%		
Smoking Status				
Yes	3392	29.4% ± 0.6%		
No	5610	70.6% ± 0.6%		
Food Insecure				
Yes	1449	14.1% ± 0.4%		
No	6158	85.9% ± 0.4%		
BMI ^2^	7607		28.7 (6.44)	13.4–76.1
Vitamin C ^3^	7607		54.4 (28.6)	0.6–274.2
CRP ^4^	7607		0.48 (0.92)	0.01–25.4
RDW ^5^	7607		12.9 (1.2)	10.7–26.9

^1^ Poverty to Income Ratio; ^2^ Body Mass Index (kg/m^2^); ^3^ Plasma vitamin C (μmol/L); ^4^ C-Reactive Protein (mg/dL); ^5^ Red Cell Distribution Width (%).

**Table 2 nutrients-14-01254-t002:** Inflammatory markers CRP and RDW across vitamin C plasma level quintiles.

	Vitamin C Plasma Level							Bonferonni Post hoc Test *p*									
	Deficiency Group I (*n* = 467)	Hypo-vitminosis Group II (*n* = 722)	Inadequate Group III (*n* = 1991)	Adequate Group IV (*n* = 2567)	Saturating Group V (*n* = 1960)	*F*	*p*	I vs. II	I vs. III	I vs. IV	I vs. V	II vs. III	II vs. IV	II vs. V	III vs. IV	III vs. V	IV vs. V
Marker	Mean ± SD	Mean ± SD	Mean ± SD	Mean ± SD	Mean ± SD												
CRP ^a^	0.67 ± 1.44	0.61 ± 1.22	0.53 ± 0.88	0.45 ± 0.91	0.37 ± 0.60	19.4	<0.001 ^c^	1.00	0.28	<0.001	<0.001	0.42	<0.001	<0.001	0.03	<0.001	0.03
RDW ^b^	13.0 ± 1.49	13.0 ± 1.35	12.8 ± 1.25	12.8 ± 1.11	12.8 ± 1.09	11.2	<0.001 ^c^	1.00	0.96	0.001	<0.001	1.00	0.001	<0.001	0.002	<0.001	1.00

^a^ C-reactive protein (mg/dL), ^b^ Red cell distribution width (%), ^c^ With Welch correction.

## Data Availability

The publicly available datasets used in this study can be found on the Centers for Disease Control and Prevention (CDC) website at: https://www.cdc.gov/nchs/nhanes/index.htm (accessed on 5 February 2022).

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
