# Peer review of "Vitamin C Plasma Levels Associated with Inflammatory Biomarkers, CRP and RDW: Results from the NHANES 2003–2006 Surveys"

_nutrients, 2022, doi:10.3390/nu14061254_

Round 1
Reviewer 1 Report
It is a well-defined and analyzed work, with a large sample size, which is a strength of the study. However, it can be improved in some respects.
INTRODUCTION
- Authors should include values indicating “Vitamin C insufficient” in page 1, line 39, and “extremely low Vitamin C”, in page 1, line 45. Furthermore, also the most commonly values to define hypovitaminosis C and vitamin C deficiency should be included (see DOI: 10.3390/nu12072008).
- Authors should explain the reason why they have chosen PCR and red cell distribution width as indicators of inflammation. I have seen a briefly explanation in “Materials and Methods” section, but it should be explained with more detail in the introduction. Furthermore, they should indicate the reason why they have not chosen other indicators such platelet cell counts or white blood cell counts, which have been related with Vitamin C levels in the blood in other studies (DOI: 10.3390/antiox7070092).
MATERIALS AND METHODS
- Have authors tested other values most commonly to define hypovitaminosis C and vitamin C deficiency instead quintiles?
- A definition, with reference, of “Food Insecure” should be included in this section.
- It would have been desirable to include information on diet, use of supplements and physical exercise, as they influence vitamin C levels in the blood.
RESULTS
- In table 1 information about supplements consumption and physical activity should be included (if authors have it).
- Are there differences in the variables showed in Table 1 among quintiles? A table showing this information should be included.
Author Response
Dear Reviewer,
We want to take this opportunity to thank you for your time in providing constructive feedback and objective questions. We hope that in the answers below and in the revised document we have been able to show that we have appreciated your perspective and edited accordingly.
Reviewer Comments:
It is a well-defined and analyzed work, with a large sample size, which is a strength of the study. However, it can be improved in some respects.
INTRODUCTION
- Authors should include values indicating “Vitamin C insufficient” in page 1, line 39, and “extremely low Vitamin C”, in page 1, line 45. Furthermore, also the most commonly values to define hypovitaminosis C and vitamin C deficiency should be included (see DOI: 10.3390/nu12072008).
Thank you for this feedback! We have edited to include the values for insufficient and extremely low plasma Vitamin C. We have included already the most commonly defined values of deficiency and hypovitaminosis but have edited for clarity.
- Authors should explain the reason why they have chosen PCR and red cell distribution width as indicators of inflammation. I have seen a briefly explanation in “Materials and Methods” section, but it should be explained with more detail in the introduction. Furthermore, they should indicate the reason why they have not chosen other indicators such platelet cell counts or white blood cell counts, which have been related with Vitamin C levels in the blood in other studies (DOI: 10.3390/antiox7070092).
This is very constructive, and we are appreciative. We have edited to elaborate more on our selection of RDW instead of WBC or PLT. We hope this revision provides more clarity as to the aim of our hypothesis.
MATERIALS AND METHODS
- Have authors tested other values most commonly to define hypovitaminosis C and vitamin C deficiency instead quintiles? We previously published a work that described the decision to examine the defined quintiles in this paper and have expanded this paper to include more of the description. Thank you very much.
- A definition, with reference, of “Food Insecure” should be included in this section. We have expanded more on the food insecure variable in the materials and methods section, thank you.
- It would have been desirable to include information on diet, use of supplements and physical exercise, as they influence vitamin C levels in the blood. This an excellent suggestion and we apologize that we did not capture these variables in this study. We will definitely keep this in mind for future works and want to thank you for providing it.
RESULTS
- In table 1 information about supplements consumption and physical activity should be included (if authors have it). Thank you for this suggestion. We apologize that we did not capture this information in this study, but agree that future endeavors should include it for more context.
- Are there differences in the variables showed in Table 1 among quintiles? A table showing this information should be included. We agree that the differences of the variables among the quintiles was extremely interesting and have included it in a previously published work. We have expanded, hopefully, enough to point readers back to this first paper. Thank you!
Very Respectfully,
Jennifer Crook, PhD, RN
Reviewer 2 Report
This observational study assesses associations between vitamin C and inflammatory markers (CRP and RDW). A major limitation of the study is that little if any of the previously published CRP/vitC data is presented and discussed. And not clear what the current study is adding to the body of knowledge over and above this.
Title –
Recommend putting NHANES in the title – may help attract more readership
‘levels are associated with the inflammatory.’
Abstract –
It is a little confusing between use of ‘associations’ and ‘differences’ – make sure you are using the correct word for what you are trying to convey. If using ‘associations’ – be clear what direction – positive or negative/inverse.
Introduction –
Since the paper is about associations of vitC with CRP and RDW – these aspects should be discussed a lot more in the introduction – as a number of studies have looked at association of vitC with CRP (there is even meta-analysis on this topic). What is your study adding to the current body of knowledge?
What was the rationale for using RDW to look at associations with vitC?
The first sentence needs to be restructured as sounds like humans lack the ability to synthesis fruit and vegetables the way it is currently written.
‘histone methylation and enzyme genesis’ should be something like ‘DNA and histone demethylation and enzyme-dependent biosynthesis of various biomolecules’.
Probably don’t need 15 references for this one sentence – should be able to cite 2-3 pertinent reviews.
Assessed for vitamin C status.
Overall, the whole paper needs to be updated for proper English grammar.
First sentence in second paragraph should be two sentences.
‘This absence of data leaves doubts in any conclusion that supplementation provides little or no benefit’. - rewrite for clarity.
‘It is possible that subclinical nutritional insufficiencies of Vitamin C partly 51 explain increased amounts of both acute and chronic inflammation as well as inadequate 52 immune responses as part of disease instigation and progression.’ – this is the most important aspect of the intro but does not have any reference to previous research in this area.
‘the purpose of this study is’ – use past tense.
Methods –
Make it clear vitC collection and analyses carried out by NHANES staff not the authors.
Move the general information about CRP and RDW into introduction. Just leave the method-specific info in the methods section.
Not sure if normal CRP assay can detect <5 mg/L accurately.
Results – Figs 1 and 2 are reproducing data from Table 2. Typically use either table or figure to represent data – not both.
Line 156 – ‘Mean levels of Vitamin C’ – should this be ‘Mean levels of CRP’?
What about analysing the data the other way round ie by standard categories of CRP and see if there is a difference in vitC levels.
Discussion -
Are the differences in CRP between deficient and saturating vitC clinically relevant?
Same with differences between RDW between lowest and highest category – is there any clinical relevance?
Line 176 – ‘Considering bioavailability of Vitamin C is fluid depending on body requirements’ – rewrite for clarity.
Line 178 – ‘Recent research’ – the research may be recent but the data being used is old – so this should be clarified as values may have changed over time.
Line 185 - ‘that number’ – what number?
Line 186 – be careful with the use of the word ‘associations’ vs differences. If using ‘associations’ – be clear what direction – positive or negative/inverse.
Line 188 – ‘mean CRP levels of < 1 mg/dL’ ie the overall range was quite large ie up to 25 mg/L.
Because study is observational only – not sure of direction of association (if it is a valid association) ie does low VitC contribute to elevated CRP or does elevated CRP contribute to lowered vitC?
Another limitation is that it is old data – values may have changed over time.
References –
#16 – author details need to be corrected.
Author Response
Dear Reviewer,
We want to take this opportunity to thank you for your time in providing constructive feedback and objective questions. We hope that in the answers below and in the revised document we have been able to show that we have appreciated your perspective and edited accordingly.
This observational study assesses associations between vitamin C and inflammatory markers (CRP and RDW). A major limitation of the study is that little if any of the previously published CRP/vitC data is presented and discussed. And not clear what the current study is adding to the body of knowledge over and above this. Thank you for this constructive feedback; we have extensively revised to include the previous literature of the association between Vitamin C and CRP and hope it is clearer how this study adds to the current body of knowledge by extrapolating the differences between the plasma quintiles (only two of which are recognized clinically at this time.)
Title –
Recommend putting NHANES in the title – may help attract more readership Thank you, we have revised the title.
‘levels are associated with the inflammatory.’
Abstract –
It is a little confusing between use of ‘associations’ and ‘differences’ – make sure you are using the correct word for what you are trying to convey. If using ‘associations’ – be clear what direction – positive or negative/inverse. Thank you for this constructive feedback. We have revised the abstract to be more clear on the differences (when appropriate) and associations identified.
Introduction –
Since the paper is about associations of vitC with CRP and RDW – these aspects should be discussed a lot more in the introduction – as a number of studies have looked at association of vitC with CRP (there is even meta-analysis on this topic). What is your study adding to the current body of knowledge? We have revised the introduction to expand the CRP and RDW rationale; thank you.
What was the rationale for using RDW to look at associations with vitC? We have included the rationale for RDW in this section now; thank you very much for the feedback.
The first sentence needs to be restructured as sounds like humans lack the ability to synthesis fruit and vegetables the way it is currently written. This is changed, thank you for the feedback.
‘histone methylation and enzyme genesis’ should be something like ‘DNA and histone demethylation and enzyme-dependent biosynthesis of various biomolecules’. Thank you.
Probably don’t need 15 references for this one sentence – should be able to cite 2-3 pertinent reviews. Revised, thank you.
Assessed for vitamin C status. Revised, thank you.
Overall, the whole paper needs to be updated for proper English grammar. Thank you for the feedback; we have revised the introduction section.
First sentence in second paragraph should be two sentences. We have significantly revised the introduction section.
‘This absence of data leaves doubts in any conclusion that supplementation provides little or no benefit’. - rewrite for clarity. We have significantly revised the introduction.
‘It is possible that subclinical nutritional insufficiencies of Vitamin C partly 51 explain increased amounts of both acute and chronic inflammation as well as inadequate 52 immune responses as part of disease instigation and progression.’ – this is the most important aspect of the intro but does not have any reference to previous research in this area. We have significantly revised the introduction, thank you.
‘the purpose of this study is’ – use past tense. Changed, thank you.
Methods –
Make it clear vitC collection and analyses carried out by NHANES staff not the authors. Revised, thank you very much.
Move the general information about CRP and RDW into introduction. Just leave the method-specific info in the methods section. Thank you for this instruction; we have revised the section.
Not sure if normal CRP assay can detect <5 mg/L accurately.
Results – Figs 1 and 2 are reproducing data from Table 2. Typically use either table or figure to represent data – not both. Thank you for this critique. We have removed the figures and left the table.
Line 156 – ‘Mean levels of Vitamin C’ – should this be ‘Mean levels of CRP’? You are correct, thank you. We have revised the sentence.
What about analysing the data the other way round ie by standard categories of CRP and see if there is a difference in vitC levels. That is an interesting concept. We will definitely keep it in mind for future endeavors.
Discussion -
Are the differences in CRP between deficient and saturating vitC clinically relevant? No, and we have made that more clear in the paper, thank you.
Same with differences between RDW between lowest and highest category – is there any clinical relevance? Same as above.
Line 176 – ‘Considering bioavailability of Vitamin C is fluid depending on body requirements’ – rewrite for clarity. Thank you for this instruction; we have revised.
Line 178 – ‘Recent research’ – the research may be recent but the data being used is old – so this should be clarified as values may have changed over time. Very true; we have revised.
Line 185 - ‘that number’ – what number? Thank you for that catch; we have revised.
Line 186 – be careful with the use of the word ‘associations’ vs differences. If using ‘associations’ – be clear what direction – positive or negative/inverse. We genuinely appreciate the feedback and have revised throughout the document.
Line 188 – ‘mean CRP levels of < 1 mg/dL’ ie the overall range was quite large ie up to 25 mg/L. Very true!
Because study is observational only – not sure of direction of association (if it is a valid association) ie does low VitC contribute to elevated CRP or does elevated CRP contribute to lowered vitC? Because it was a cross sectional study, we aren’t sure that we were able to answer that question with this data. However, it would be interesting to study in the future.
Another limitation is that it is old data – values may have changed over time. We agree that the prevalence values may have changed over time, but the association identified should continue to be tested to either confirm or disprove what we found in this study.
References –
#16 – author details need to be corrected. We greatly apologize for that and have revised.
Very Respectfully,
Jennifer Crook, PhD, RN
Reviewer 3 Report
While this manuscript is on an interesting topic, and the results of the analysis seem to match with what is expected with vitamin C and inflammation, there are some shortcomings that should be addressed before publication. In particular, some of the statements made by the authors are not supported by previous studies. In some cases, there are extensive grammatical errors that make interpretation difficult. Lastly, the definition of vitamin C inadequacy and hypovitaminosis C does not match previous reports in this field, making it difficult to match up with other work. Thus, revisions are suggested. Specific comments follow.
- Vitamin C should not be capitalized unless it starts a sentence.
- Lines 28-29: As written, it sounds like humans can synthesize fruits and vegetables. Revise for clarity.
- Line 32: What is "enzyme genesis"?
- Line 33: The effect on COVID is specifically intravenous vitamin C. This should be added for clarification.
- Lines 37-41: Revise for clarity. This is a run-on sentence.
- Line 45: Reference 23 only mentioned COVID-19 and sepsis passively (Sepsis is only mentioned once in the entire article). If a claim is to be made about the effect of these conditions on vitamin C, it should link to the actual data.
- Lines 48-50: This is poorly written. Increased supplementation from what level?
- Lines 61-64: These categories (of vitamin C status) are arbitrary and not in line with other publications on this topic. If this is not revised to be in line with other papers, the differences (and explanation of them) should be at least acknowledged in the paper somewhere.
- Line 79: How and when was blood collected? How was it stored? How was it prepared?
- Line 80-82: More details are needed. What was the external standard (and in what concentrations)? What was the stationary and mobile phase? What is the retention time? These details are completely missing from the manuscript.
- Similar to #10 - how was CRP and RDW assessed? What measurements were used? What lab did the work?
- Table 1: Is the vitamin C represented here plasma values or dietary intake levels? Since dietary intake levels are available through NHANES, this should also be noted here.
- Related to #11 - fasting plasma values of vitamin C should rarely above 100 uM. Are these fasting blood draws? If not, that makes the interpretation very difficult. How many blood vitamin C values exceeded 80uM throughout the entire dataset?
- Figure 1 needs error bars. It is unnecessary to provide the values, but if it is kept on the figure, they should have better contrast with the blue bars. What would be more valuable for the data in Figure 1 would be an analysis of the distribution of CRP values - box and whisker plots instead of means.
- Line 176: There is no evidence presented here (or in the background materials) that the bioavailability of vitamin C is fluid. This requires some evidence in support.
- Throughout the manuscript: There is no reason to use 4 significant figures in all dataset.
Author Response
Dear Reviewer,
We want to take this opportunity to thank you for your time in providing constructive feedback and objective questions. We hope that in the answers below and in the revised document we have been able to show that we have appreciated your perspective and edited accordingly.
While this manuscript is on an interesting topic, and the results of the analysis seem to match with what is expected with vitamin C and inflammation, there are some shortcomings that should be addressed before publication. In particular, some of the statements made by the authors are not supported by previous studies. In some cases, there are extensive grammatical errors that make interpretation difficult. Lastly, the definition of vitamin C inadequacy and hypovitaminosis C does not match previous reports in this field, making it difficult to match up with other work. Thus, revisions are suggested. Specific comments follow.
- Vitamin C should not be capitalized unless it starts a sentence. Thank you for this correction; we have revised throughout the manuscript.
- Lines 28-29: As written, it sounds like humans can synthesize fruits and vegetables. Revise for clarity. We have revised the sentence.
- Line 32: What is "enzyme genesis"? The creation of enzymes; we have rewritten the statement and hope this is clearer.
- Line 33: The effect on COVID is specifically intravenous vitamin C. This should be added for clarification. Thank you for that point; we have revised the statement.
- Lines 37-41: Revise for clarity. This is a run-on sentence. Thank you; we have revised.
- Line 45: Reference 23 only mentioned COVID-19 and sepsis passively (Sepsis is only mentioned once in the entire article). If a claim is to be made about the effect of these conditions on vitamin C, it should link to the actual data. We have included another reference regarding supplementation of vitamin C in patients with sepsis.
- Lines 48-50: This is poorly written. Increased supplementation from what level? We have revised the sentence for clarity.
- Lines 61-64: These categories (of vitamin C status) are arbitrary and not in line with other publications on this topic. If this is not revised to be in line with other papers, the differences (and explanation of them) should be at least acknowledged in the paper somewhere. Thank you; we have included the rationale for the ranges in lines 115-120.
- Line 79: How and when was blood collected? How was it stored? How was it prepared? We have included clearer details in lines 97-99.
- Line 80-82: More details are needed. What was the external standard (and in what concentrations)? What was the stationary and mobile phase? What is the retention time? These details are completely missing from the manuscript. Thank you; we have included more description in lines 108-112.
- Similar to #10 - how was CRP and RDW assessed? What measurements were used? What lab did the work? We have revised the sections from lines 97-130.
- Table 1: Is the vitamin C represented here plasma values or dietary intake levels? Since dietary intake levels are available through NHANES, this should also be noted here. These are plasma values. This was part of a larger work. We have included in our first paper the dietary intake values, however, they were not found to be associated with plasma values and so we did not include them in this manuscript but we did provide more of that information in lines 45-48.
- Related to #11 - fasting plasma values of vitamin C should rarely above 100 uM. Are these fasting blood draws? If not, that makes the interpretation very difficult. How many blood vitamin C values exceeded 80uM throughout the entire dataset? These are not fasting values. The amount of participants with values >80 are approximately 600. We have included this as a limitation; thank you for the feedback.
- Figure 1 needs error bars. It is unnecessary to provide the values, but if it is kept on the figure, they should have better contrast with the blue bars. What would be more valuable for the data in Figure 1 would be an analysis of the distribution of CRP values - box and whisker plots instead of means. We have removed the figures and left the table.
- Line 176: There is no evidence presented here (or in the background materials) that the bioavailability of vitamin C is fluid. This requires some evidence in support. We have revised the statement.
- Throughout the manuscript: There is no reason to use 4 significant figures in all dataset. It is unclear what you mean by this.
Very Respectfully,
Jennifer Crook, PhD, RN

Round 2
Reviewer 1 Report
The article has been changed and improved. However, authors should include in the limitations of the study that information about physical activity has not been included (since this influences the levels of vitamin C).
Author Response
March 8, 2022
Dear Reviewer,
We want to thank you again for your constructive feedback. We have edited as suggested and hope our revisions address more clearly the item pointed out.
The article has been changed and improved. However, authors should include in the limitations of the study that information about physical activity has not been included (since this influences the levels of vitamin C).
Thank you for this suggestion; we have included it. Unfortunately, the NHANES does not capture physical exercise well, but we agree that it is a factor that must be addressed and will be remembered for future research endeavors.
Again, we are extremely grateful for your thoughtful review!
Very Respectfully,
Jennifer Crook
Reviewer 2 Report
The authors have addressed a majority of my suggestions. A few minor points:
Line 31-35. I still think 10 references is more than is needed for this one sentence. It is more important to include references later that are directly related to the biomarkers of interest.
Line 61-63. ‘extensive literature’ has one case study (ref 20) cited. There is meta-analysis available.
Line 66. strong predictor of risk of developing cardiovascular disease
Author Response
March 8, 2022
Dear Reviewer,
We want to thank you again for your constructive feedback. We have edited as suggested and hope our revisions address more clearly the items pointed out.
The authors have addressed a majority of my suggestions. A few minor points:
Line 31-35. I still think 10 references is more than is needed for this one sentence. It is more important to include references later that are directly related to the biomarkers of interest. Thank you for this excellent suggestion; we have refined the references in both the introduction paragraph as well as the biomarkers’ paragraphs.
Line 61-63. ‘extensive literature’ has one case study (ref 20) cited. There is meta-analysis available. We have included more evidence, including meta-analysis, thank you.
Line 66. strong predictor of risk of developing cardiovascular disease. We have edited the phrase.
Again, we are extremely grateful for your thoughtful review!
Very Respectfully,
Jennifer Crook
Reviewer 3 Report
Overall, this manuscript supports the relationship between plasma vitamin C levels and a limited set of inflammatory markers. It is improved from the original version, but some changes are still suggested:
In response to the author's reply:
- It was not suggested that Figure 1 (from original submission) should be removed - this was a good addition to the manuscript. However, how it was presented could have been improved. A box-and-whisker plot would have been highly informative.
- Significant figures are typically limited to 2 or 3 (as per journal guidelines). Typically four significant figures in data are considered useless and are usually not borne out by the precision of the assays involved. Indeed, the assays presented here cannot do not support such precision (i.e., the extra significant figures are well within the error of the assay and should be discarded).
Other issues that remain:
- Line 30: Grammar issue: Vitamin C cannot rely on anything. However, stating vitamin C levels or status would be appropriate.
- Line 45: Suggest pushing this sentence (starting with "However, patients...") into the next paragraph as it changes the topic and presents a persistent issue for vitamin C research.
- Line 50: Hypovitaminosis C has no standard clinical definition. It might be better just to remove the word clinical or refer to it differently.
- Line 102: The reference here does not provide any helpful information in the NHANES blood collection. It would be better to reference a more informative document, such as https://www.cdc.gov/nchs/data/series/sr_02/sr02_170.pdf
- Line 108: What is unclear from this paper is if the authors performed the analyses themselves, or if they acquired the data obtained by the NHANES group. If acquired, the description of the analysis can be kept minimal and references to the published description would suffice (there are PDF reference documents for the vitamin C procedure available on the NHANES site, for example). If the authors did perform the analysis, more description is needed about how their procedure differed from published protocols (if any difference was notable).
- Spelling, grammar, and punctuation issues remain.
Author Response
March 8, 2022
Dear Reviewer,
We want to thank you again for your constructive feedback. We have edited as suggested and hope our revisions address more clearly the items pointed out.
Overall, this manuscript supports the relationship between plasma vitamin C levels and a limited set of inflammatory markers. It is improved from the original version, but some changes are still suggested:
In response to the author's reply:
- It was not suggested that Figure 1 (from original submission) should be removed - this was a good addition to the manuscript. However, how it was presented could have been improved. A box-and-whisker plot would have been highly informative. We removed the figures on the advice of another reviewer. We are including a modified version as a box-and-whisker plot and hopes it is sufficient. Thank you for the added feedback on this.
- Significant figures are typically limited to 2 or 3 (as per journal guidelines). Typically four significant figures in data are considered useless and are usually not borne out by the precision of the assays involved. Indeed, the assays presented here cannot do not support such precision (i.e., the extra significant figures are well within the error of the assay and should be discarded). Thank you very much for the explanation. We have revised the data throughout the manuscript to be limited to 3 significant figures. We are grateful for the feedback.
Other issues that remain:
- Line 30: Grammar issue: Vitamin C cannot rely on anything. However, stating vitamin C levels or status would be appropriate. Thank you for that catch; we have revised the sentence.
- Line 45: Suggest pushing this sentence (starting with "However, patients...") into the next paragraph as it changes the topic and presents a persistent issue for vitamin C research. That does make more sense, thank you for the suggestion.
- Line 50: Hypovitaminosis C has no standard clinical definition. It might be better just to remove the word clinical or refer to it differently. We have removed the word clinical, thank you.
- Line 102: The reference here does not provide any helpful information in the NHANES blood collection. It would be better to reference a more informative document, such as https://www.cdc.gov/nchs/data/series/sr_02/sr02_170.pdf Thank you for this; we have revised the section.
- Line 108: What is unclear from this paper is if the authors performed the analyses themselves, or if they acquired the data obtained by the NHANES group. If acquired, the description of the analysis can be kept minimal and references to the published description would suffice (there are PDF reference documents for the vitamin C procedure available on the NHANES site, for example). If the authors did perform the analysis, more description is needed about how their procedure differed from published protocols (if any difference was notable). We tried to clarify further that the specimen collection and analysis was conducted by the NHANES and not the authors. We hope this revision makes this point stronger.
- Spelling, grammar, and punctuation issues remain. We have corrected, thank you.
Again, we are extremely grateful for your thoughtful review!
Very Respectfully,
Jennifer Crook